# Research Progress on the Effect and Mechanism of Superchilling Preservation Technology on Meat Quality Control

**DOI:** 10.3390/foods13203309

**Published:** 2024-10-18

**Authors:** Bo Wang, Jiamin Liang, Changyu Zhou, Jiamin Zhang, Lili Ji, Congyan Li, Xiuli Mei, Hongyue Chen

**Affiliations:** 1Meat Processing Key Laboratory of Sichuan Province, College of Food and Biological Engineering, Chengdu University, Chengdu 610106, China; wangbo9214@163.com (B.W.); ljm13672537719@126.com (J.L.); jilili@cdu.edu.cn (L.J.); 2Key Laboratory of Animal Protein Food Processing Technology of Zhejiang Province, College of Food Science & Engineering, Ningbo University, Ningbo 315211, China; zhouchangyu@nbu.edu.cn; 3Sichuan Academy of Animal Husbandry Science, Chengdu 610106, China; licongyan0925@163.com (C.L.); meixiuli0716@163.com (X.M.); 4Chongqing Animal Husbandry and Veterinary Technology Extension Station, Chongqing 710014, China; chy106@163.com

**Keywords:** superchilling, preservation technologies, electric fields, magnetic fields, meat quality

## Abstract

During storage and transportation, meat is susceptible to the effects of microorganisms, endogenous enzymes, and oxygen, leading to issues such as moisture loss, spoilage, and deterioration. Superchilling, as a preservation method that combines the benefits of refrigeration and freezing, can effectively slow the growth and reproduction of microorganisms, control protein and lipid oxidation, reduce water loss, and maintain the quality and sensory properties of meat. This paper reviews the current application status of superchilling technology in meat preservation, focusing on the mechanisms of ice crystal formation, water retention, tenderness preservation, protein and fat oxidation control, and microbial growth inhibition under superchilling conditions. Additionally, it summarizes the research progress on the combined application of superchilling with emerging technologies such as electric fields, magnetic fields, and electron beams in meat preservation and explores its potential and future prospects for improving meat quality. The aim is to provide scientific evidence and technical support for the application of superchilling technology in enhancing meat quality.

## 1. Introduction

Meat is a crucial source of nutrition for humans, providing abundant high-quality proteins, essential amino acids, fats, vitamins, and minerals. With the improvement in living standards, meat has become increasingly popular among consumers due to its diverse textures and flavors [1]. However, during storage and transportation, meat is highly susceptible to spoilage caused by factors such as microbial activity, endogenous enzymes, and oxygen exposure [2]. This not only contributes to food waste but also heightens the risk of foodborne illnesses resulting from meat spoilage and pathogenic bacteria [3].

To improve the storage quality of meat and extend its shelf life, low-temperature preservation has become the most commonly used technology for meat storage due to its relatively low damage to the structure and appearance of meat [4]. Superchilling, as a preservation method that combines the advantages of refrigeration and low-temperature freezing (<−8 °C), has attracted significant attention and research. Superchilling preservation refers to a method of preserving meat by placing it in a temperature range of 1–2 °C below its freezing point. During the superchilling process, meat is in a partially frozen state, where the activities of enzymes and microorganisms are limited, and moisture freezes into ice, depriving microorganisms of a suitable growth environment, thus inhibiting enzyme activity and the vitality of microbial growth and reproduction [5]. Compared to refrigeration, superchilling preservation has lower storage temperatures, better antioxidant effects, and less drip loss and quality loss [6]. Compared to freezing, superchilling can maintain good muscle microstructure. With the continuous development of superchilling technology, its research and application in meat processing and storage are becoming increasingly widespread. However, there is currently a lack of reports elucidating the mechanisms underlying superchilling preservation in meat products.

This paper reviews the current application status of superchilling preservation technology in meat, focusing on the mechanisms of ice crystal formation in meat under superchilling conditions, water retention mechanisms, tenderness retention mechanisms, control mechanisms of protein and fat oxidation, and the inhibition of microbial growth. It also summarizes the research progress on the composite application of superchilling with novel technologies such as electric fields, magnetic fields, and electron beams for meat preservation, and discusses its potential and prospects for improving meat quality. The aim is to reveal how superchilling regulates meat quality to achieve preservation effects, providing a scientific basis and technical support for the application of superchilling technology in enhancing meat quality, and offering more scientific storage methods and technical support for the meat processing industry, thereby promoting the healthy development of the meat industry.

## 2. Superchilling Preservation Technology

### 2.1. Principles of Superchilling Technology

During the superchilling process, only a portion of the total water in the meat is frozen—the content of the generated ice crystals is about 5–30% [7]. The physiological and biochemical reactions of endogenous enzymes are inhibited, and some microorganisms are also suppressed or even killed due to the changed environment. This poses a serious threat to the growth and reproduction activities of microorganisms, thereby slowing down the rigor mortis and resolution periods of the meat, which in turn reduces the rate of spoilage, allowing the meat to maintain good quality for a certain period and extending the shelf life of the food. In addition, small ice crystals are uniformly distributed within the meat, reducing damage to the tissue structure and effectively maintaining its original freshness. Compared to refrigeration, the shelf life of meat after superchilling can be extended by at least 1.4 to 4 times [8]. Although the shelf life of superchilled meat is far less than that of low-temperature freezing (<−8 °C), the mechanical damage caused by superchilling is less than that caused by low-temperature freezing. Consequently, the mechanical damage to muscle/meat structure caused by ice crystal is less than that obtained from low-temperature freezing (<−8 °C), which helps reduce moisture loss during storage.

### 2.2. The Preservation Effect of Superchilling Technology in Meat

The concept of superchilling was first proposed by the Frenchman Le Danois in 1920 [9]. Initially applied in the fishery industry, it has gradually expanded to include the preservation of poultry and livestock meat, as well as fruits and vegetables, with the deepening of research. Currently, superchilling technology is widely utilized in the preservation of meat, including poultry (e.g., chicken and duck), livestock (e.g., pork and lamb), and seafood (e.g., fish and shrimp) [10]. The application effects of superchilling technology in meat preservation are primarily evaluated through indicators such as pH, TBARS (thiobarbituric acid reactive substance), TVB-N (total volatile basic nitrogen), K value, and total bacterial count, as shown in Table 1.

Through Table 1, it can be seen that compared to traditional refrigeration and freezing techniques, superchilling inhibits the growth and reproduction of microorganisms, delays the oxidation processes of meat proteins and lipids, and maintains desirable texture, color, and water-holding capacity. In the application of seafood, superchilling technology significantly reduces the K value and minimizes the formation of biogenic amines, which is more beneficial for preserving freshness. Additionally, during superchilling, the formation of ice on the surface creates a protective layer that prevents adverse effects from rapid external temperature fluctuations. During transportation, even with temperature variations, this surface ice provides protection, helping to maintain the internal temperature and thereby preserving quality and freshness [9].

As shown in Table 1, the superchilling temperatures for poultry, livestock, and seafood products are between −1 °C and −5 °C. After superchilling, the meat demonstrates reduced drip and juice loss, indicating improved water retention capacity. Furthermore, superchilled meat exhibits lower TBARS and TVB-N values, along with a slower oxidation process of actomyosin and myoglobin. This method effectively slows the oxidation of proteins and fats, mitigates the increase in the K value, and decreases the formation of biogenic amines, thereby better preserving freshness. Additionally, the reduced total bacterial count and the slower pH decline indicate that microbial growth in meat is effectively inhibited after superchilling. Moreover, superchilling slows down the consumption of glucose and the post-mortem glycolysis process in meat, maintaining a higher pH value. Both color and sensory scores remain stable, and the structure of the muscle tissue is preserved relatively intact. In terms of shelf life, superchilling effectively extends the shelf life of meat while maintaining good quality and freshness. Xiao et al. [20] found that the shelf life of yellow cattle tripe and water buffalo tripe stored using superchilling (6 days) was doubled compared to refrigeration. Research by Lee et al. [15] indicated that beef stored at 3 °C for 14 days exceeded the consumer safety standard of 7 colony-forming units per gram, while beef stored at −2 °C under superchilling conditions remained below 7 cfu/g, effectively limiting microbial growth and extending the beef’s shelf life by at least twofold. Chan et al. [21] found that Atlantic salmon stored under modified atmosphere refrigeration was completely spoiled after 14 days, whereas Atlantic salmon stored under modified atmosphere superchilling was only deemed spoiled after 49 days, demonstrating a longer shelf life for the superchilling group compared to the control group.

## 3. Mechanisms of Superchilling Technology in Regulating Meat Quality

### 3.1. The Mechanism of Ice Crystal Formation in Superchilling

The mode of ice crystal formation and the resulting ice crystal morphology during the superchilling of meat are key factors in regulating meat quality. Ice crystal formation is a typical thermodynamic physical phenomenon that reflects the phase transition of water into ice. During this process, water molecules in food, under a specific degree of supercooling, crystallize and release latent heat, gradually arranging themselves into a solid lattice structure [22]. As shown in Figure 1, the formation of ice crystals during the superchilling of meat primarily involves the following four stages [23]: (1) Supercooling stage: During superchilling, as the temperature decreases, the internal temperature of the food reaches the freezing point of water but has not yet reached the temperature required for crystallization. This state, where the temperature is below the freezing point of water but crystallization has not yet occurred, is referred to as the supercooling state. (2) Ice nucleation: A specific degree of supercooling provides the driving force for water molecules in the food to aggregate and form an initial ice crystal nucleus. Once a certain volume is reached, a stable nucleus is gradually formed. (3) Ice crystal growth: After a stable nucleus is formed, water molecules in the food orderly accumulate on the surface of the nucleus to form small ice crystals, which grow rapidly in different directions along the nucleus. (4) Recrystallization: Due to vapor pressure, smaller ice crystals tend to merge with larger ones. Crystal fusion or temperature fluctuations lead to recrystallization, where changes in size, number, or shape occur without altering the overall mass of ice crystals. Controlling the nucleation and growth of ice crystals is a critical approach to improving the quality of superchilled meat. By regulating these two stages, it is possible to effectively promote the uniform distribution of ice crystals, control their size and morphology, and maintain stable quality. Ice crystal nucleation can be divided into homogeneous nucleation and heterogeneous nucleation. Homogeneous nucleation refers to the spontaneous aggregation of water molecules in solution to form crystal nuclei, while heterogeneous nucleation occurs when water forms nuclei on dust particles, the surface of food containers, or micro-heterogeneous substances within food [24]. In superchilled meat, heterogeneous nucleation is more common than homogeneous nucleation due to the presence of various heterogeneous substances within the meat, such as fats and proteins, which serve as nucleation points for ice crystals, facilitating their formation. Homogeneous nucleation occurs only in pure liquids. Rajib Lochan Poudyal et al. [25] found that ice crystal nucleation is a random phenomenon, with the location and temperature of nucleation varying between samples, even when the storage conditions and internal composition of the samples are similar. Kobayashi et al. [26] demonstrated that salmon undergoes heterogeneous nucleation during superchilling, regardless of temperature changes, due to its complex muscle structure containing a substantial amount of fat and protein.

### 3.2. The Mechanism of Water Retention

Water-holding capacity refers to the ability of muscle to retain its intrinsic moisture and absorb additional water when subjected to external forces such as heating, pressure, and processing. It is commonly evaluated through indicators such as cooking loss, thawing loss, centrifugation loss, and drip loss [27]. The structure of muscle fiber is shown in Figure 2. The water-holding capacity of meat during freezing may be affected by the following two factors. First, large ice crystals can damage the integrity of myofibrils and muscle cells. During thawing, the tightly bound water (immobilized water) within the muscle fibers is displaced to the extracellular space, becoming free water. This free water, along with certain nutrients, is lost through the damaged muscle tissue, thereby reducing the moisture content of the thawed meat. Superchilling can inhibit the formation and growth of ice crystals, reducing the number of large ice crystals and limiting the extent of damage to myofibrils and muscle cells. Consequently, more immobilized water remains bound within the muscle fibers, decreasing its conversion to free water (water in the extracellular space) and minimizing the loss of free water from the meat [28]. Park et al. [29] discovered that superchilling technology, by slowing the formation and growth of ice crystals in meat, minimizes the damage caused by ice crystals to muscle cells. This process helps to preserve the integrity of muscle fibers, particularly the Z-line, M-line, and H-line, thereby improving the water-holding capacity of meat after superchilling. Secondly, the water-holding capacity of meat is closely related to the functional properties of proteins, such as solubility, emulsification, and gelation, as well as surface charge and sulfhydryl content. The growth of ice crystals induces the leakage of pro-oxidative substances from cells, which can lead to protein denaturation, a decrease in functional properties, and alterations in surface charge and sulfhydryl content. These changes compromise the stability of the protein’s three-dimensional network, reducing its ability to retain immobilized water and resulting in an increase in free water within the meat [30]. Fu et al. [31] found through comparative studies that the myofibrillar protein content, solubility, total sulfhydryl content, and active sulfhydryl content in the refrigerated group decreased more rapidly than in the superchilled group. Additionally, the emulsifying ability of myofibrillar proteins in the refrigerated group was lower than that in the superchilled group during the later stages of storage. Based on the changes in the relaxation peak area of chicken breast meat, superchilling was found to be more effective than conventional refrigeration in preserving immobilized water and reducing water loss during storage. This effect is mainly due to the slower degradation and denaturation of myofibrillar proteins in superchilled storage, which enhances the capacity to retain immobilized water, thereby inhibiting moisture loss from the meat. Consequently, superchilling can protect the functional properties and structural stability of proteins by controlling the formation of large ice crystals, ultimately reducing water loss.

In addition, compared to low-temperature freezing (<−8 °C), superchilling only partially freezes the water content, leading to more gradual temperature changes and a smaller temperature gradient between the surface and the interior of the meat. This reduces the driving force for moisture migration, allowing for slower water movement and consequently more water retention inside the cells. Due to the relatively higher temperatures of superchilling, surface moisture in the meat exists mainly in the form of liquid water or small ice crystals, whereas during low-temperature freezing (<−8 °C), surface water molecules sublimate into vapor, resulting in an overall reduction in moisture content in frozen meat, with particularly significant loss of surface moisture. This view is supported by the findings of Rathod et al. [32], who confirmed that meat stored under superchilling conditions had higher a moisture content, which was more favorable for water retention. The migration of moisture led to a significantly lower percentage of water content in the frozen group compared to the superchilled group in chicken breast samples.

### 3.3. The Mechanism of Tenderness Improvement

Tenderness is one of the most important indicators for evaluating meat quality. It refers to the texture of the meat and its products during consumption, including the ease of cutting and chewing, which is generally measured by shear force [33]. Three major factors affecting meat tenderness are the content of connective tissue, sarcomere length, and the degree of myofibrillar protein degradation by enzymes [34]. The amount of connective tissue determines the baseline toughness of the meat and is directly related to overall hardness and chewiness. Meat with higher connective tissue content tends to be tougher, while lower content results in a more tender texture [35]. Sarcomere length, an intrinsic indicator of muscle tenderness, also plays a key role. The longer the sarcomere, the more tender the meat. During rigor mortis, sarcomere length shortens, leading to tougher meat [36]. Therefore, delaying the onset of rigor mortis can effectively improve postmortem meat tenderness. Research by Yang [37] demonstrated that superchilling can lead to a sustained decrease in Ca^2+^-ATPase activity, thereby slowing metabolism and ATP depletion in muscle. This delay in the onset of rigor mortis prevents rapid contraction of the sarcomeres, ultimately improving tenderness [38,39]. In addition, the rapid freezing of postmortem muscle at low temperatures causes the rapid contraction of muscle fibers. Even after the maturation process, the meat does not soften, which increases the hardness of the meat, a phenomenon known as cold shortening. Muscle tissue is strongly stimulated by low temperatures, resulting in the inability of the sarcoplasmic reticulum to maintain normal function. A large amount of Ca^2+^ is released from the sarcoplasmic reticulum, and the rapidly rising Ca^2+^ levels in the sarcoplasm activate ATPase, which increases the amount of myosin and actin binding, resulting in excessive contraction of the muscle [40]. The milder superchilling temperature may reduce the degree of damage to the sarcoplasmic reticulum and the amount of Ca^2+^ remaining in the sarcoplasmic reticulum, thereby reducing the binding of actin and myosin [39]. In addition, the superchilling process can lead to a sustained decrease in Ca^2+^-ATPase activity, which may also reduce the production of actomyosin and reduce muscle hardness.

Moderate myofibrillar proteolysis can extend the maturation time of meat, which is a key factor in improving meat tenderness [40]. In the later stages of storage, superchilling technology can effectively reduce the activity of endogenous enzymes in muscle, particularly calpains, by slowing their reaction rates and the movement of calcium ions. This deceleration allows the enzymatic degradation process to occur more slowly, thus prolonging the maturation time of the meat. As a result, the degradation of proteins and muscle fibers is more gradual and sustained, enhancing the tenderness of the meat. Bao et al. [34] found that μ-calpain and m-calpain within the calpain system play significant roles in meat tenderization during muscle contraction and maturation by hydrolyzing structural proteins in the muscle. These enzymes are critical proteolytic agents in the meat aging process. Additionally, calpains gradually hydrolyze myosin and troponin-T, which reduces the strength of muscle fibers and decreases meat rigidity. This action further prevents the texture hardening of meat due to cold shrinkage during freezing, resulting in a more uniform tenderization throughout the maturation process. Di et al. [41] indicated that structural proteins such as myosin and troponin-T play a positive role in muscle contraction, and their degradation facilitates the elongation of sarcomeres and enhances tenderness. Chen et al. [42] found that during the storage of bone-in longissimus dorsi from beef, the superchilling group exhibited the lowest integrity of myosin and the highest level of troponin degradation, providing evidence for the superior tenderness achieved under superchilling conditions. Furthermore, although superchilling preserves a more intact microstructure compared to conventional freezing, enzymatic reactions during the maturation of meat can lead to some degree of disruption of the Z-lines within the sarcomeres. This results in a gradual elongation of the sarcomeres, contributing to the tenderization of the meat that remains in a rigor state. Research has shown that adequate moisture is also beneficial for maintaining the normal structure of the sarcomeres, preventing excessive shrinkage due to dehydration, which in turn helps to retain the tenderness of the meat [43].

### 3.4. The Mechanism of Controlling Oxidation

Oxidation of proteins and lipids can lead to flavor deterioration, dull color, decreased texture quality, and reduced nutritional value in meat and meat products. Protein and lipid oxidation occur almost simultaneously during the storage of meat, affecting and interacting with each other [44]. As illustrated in Figure 3, oxidation in meat occurs through a chain reaction involving free radicals. Protein oxidation can occur directly via reactions with reactive free radicals, while byproducts generated from lipid oxidation can indirectly induce covalent modifications of proteins [45]. Lipid oxidation can be categorized based on the oxidative pathway into autoxidation, photosensitized oxidation, and enzymatic oxidation, with autoxidation induced by free radical chain reactions being the primary pathway [46]. In addition, hemoglobin can accelerate lipid oxidation as a prooxidant by binding oxygen and promoting the production of reactive oxygen species (ROS). Meanwhile, methemoglobin is generated during the automatic oxidation of hemoglobin, and the released heme and iron ions can catalyze lipid oxidation.

Superchilling storage inhibits the activity of reactive oxygen species and oxidizing enzymes, scavenges free radicals, and chelates metal ions. It enhances the activity of antioxidant enzymes, such as superoxide dismutase, glutathione peroxidase, and catalase. Additionally, superchilling promotes the generation of antioxidant compounds, such as polyphenols, and enhances their stability, which is beneficial for slowing down oxidative processes in meat during storage. Zheng et al. [17] categorized Largemouth Bass into three groups based on storage temperature: refrigeration, superchilling, and refrigeration combined with superchilling. During storage, the overall content of actin in Largemouth Bass showed a declining trend. However, compared to the superchilling group, the actin concentration in the refrigeration group and the refrigeration plus superchilling group decreased at a faster rate, with more severe denaturation of actin. This indicates that superchilling storage, to some extent, slows down the oxidative denaturation of actin. Actin is a crucial component of myofibrillar proteins, and changes in actin concentration can reflect the degree of oxidative denaturation of meat myofibrillar proteins to some extent [47]. Furthermore, the formation of high-iron myoglobin and the exposure of heme iron can also indicate the extent of oxidation in meat. Wang et al. [48] found that, compared to superchilling storage, rabbit meat stored under refrigeration conditions showed a significant increase in the ratio of thiobarbituric acid reactive substances (TBARS) to high-iron myoglobin, along with a notable decrease in extractable heme iron content and a significant increase in non-heme iron content. Additionally, the protein carbonyl content increased significantly while the thiol content decreased significantly, indicating that superchilling storage effectively slows down the oxidation process. The study also revealed that lipid oxidation and protein oxidation in rabbit meat occur simultaneously and influence each other. Zhao et al. [49] discovered that under superchilling storage conditions, the activities of three types of proteases in the sarcoplasm of sturgeon were higher than those in refrigeration storage, suggesting that superchilling storage can effectively delay the oxidative degradation of proteins.

### 3.5. The Mechanism of Antimicrobial Action

Microbial growth and reproduction are key factors contributing to the spoilage of meat. The high moisture and nutrient content in meat create a conducive environment for microbial proliferation, while storage temperature significantly influences the development of spoilage microorganisms. During the superchilling of meat, a portion of the moisture is frozen into ice, which partially inhibits microbial growth. Furthermore, the metabolic activities and ATP production of microorganisms in superchilled meat are altered, subsequently affecting their growth. Research has shown that ATPase is an important protease involved in microbial energy metabolism and substance transport, essential for maintaining normal physiological functions in microorganisms [50]. Qin et al. [19] found that during superchilling storage, the ATP and GTP metabolic pathways in shrimp underwent specific transformations, with the low-temperature environment altering the degradation pathways of nucleotides. GMP was not detected in the samples from the refrigerated group, while the levels of GMP in the superchilled shrimp increased in the early storage period before gradually degrading. This indicates that superchilling effectively inhibits the degradation process of shrimp, leading to HX production and thereby suppressing its physiological metabolic activity.

The microbial composition of meat is complex. However, in a lower-temperature environment, only a few dominant psychrotrophic bacteria participate in the spoilage process and produce spoilage metabolites, such as Pseudomonas, Enterobacter, Staphylococcus, and Brochothrix thermosphacta [51]. Superchilling can inhibit the growth and reproduction of dominant bacteria in meat, effectively slowing down microbial growth and reproduction. Liang et al. [52] studied the effects of superchilling at −4 °C, refrigeration, and room temperature storage on the bacterial population changes in lamb meat. The results indicated that the growth and reproduction of most bacteria were inhibited under superchilling at −4 °C. Although Pseudomonas was identified as the dominant spoilage bacterium, its population in the superchilled samples was significantly suppressed. The study also confirmed that the potential for microbial-induced spoilage in meat is related to the ability of microorganisms to produce spoilage metabolites. It was found that the metabolic abundance of bacteria from lamb meat in the refrigeration group began to increase after one day of storage, whereas the superchilled group showed the opposite trend, with a decrease in the abundance of both metabolic pathways. This suggests that superchilling storage can effectively reduce the metabolic activity of bacteria concerning carbohydrates and amino acids, inhibiting microbial-induced spoilage of meat. This finding aligns with the results of Zhao et al. [53]. Additionally, the results from Duan et al. [54] also indicated that compared to the refrigeration group, the total bacterial count and diversity in superchilled tilapia fillets were lower, demonstrating that superchilling effectively inhibits microbial growth and reproduction.

## 4. Superchilling Composite Novel Preservation Technology

While superchilling effectively preserves meat quality and extends its shelf life, relying solely on this technique for meat storage presents certain limitations. For instance, some moisture can still freeze into irregular ice crystals, potentially causing mechanical damage to cells and muscle tissue. Moreover, the inhibitory effect on certain psychrotrophic bacteria may not be adequate. Additionally, superchilling demands precise temperature control. Fluctuations in temperature and variations in cooling rates can result in recrystallization of the meat [2]. Therefore, the incorporation of novel techniques such as electric fields, magnetic fields, or electron beams alongside superchilling can improve meat preservation outcomes.

### 4.1. Composite Preservation Application of Superchilling and Electric Fields

The combined use of superchilling and electric fields, recognized as an emerging preservation technology, operates through several key mechanisms: (1) The external electric field influences the potential difference across the biological membrane, affecting the electron transfer involved in redox reactions within the organism and subsequently altering its biochemical processes [54]. (2) The electric field impacts the transmembrane potential of cells, as changes in membrane potential affect the permeability of the cell membrane and thus influence cellular biochemical reactions [55]. (3) Low-frequency electrostatic waves emitted from the discharge plate affect the distribution and activity of moisture in meat, altering the interaction between water and enzymes, effectively extending shelf life and delaying spoilage processes [56]. The main types of electric fields include high-voltage static electric fields, low-voltage static electric fields, high-voltage alternating electric fields, and pulsed electric fields. Static electric fields are defined as steady-state electric fields that remain constant over time. High-voltage static electric fields typically refer to direct current voltages exceeding 1200 V, although the specific electric field strength is not clearly defined [57]. Low-voltage static electric fields are characterized by stable low-voltage electric fields created using voltages generally less than 2500 V and weak currents under 0.2 mA [58]. High-voltage alternating electric fields leverage microwave effects to inhibit ice crystal formation and cellular metabolic activities. Pulsed electric fields generate high-intensity electric fields between two electrodes within the processing chamber, with pulse amplitudes ranging from 100 to 300 V·cm^−1^ and 20 to 80 kV·cm^−1^. These fields act on products for brief periods to inactivate microorganisms [59]. The electric field primarily assists superchilling by influencing the formation of ice crystals and affecting the biochemical reactions within the biological organism. The electric field alters the arrangement and distribution of water molecules and their crystallization behavior, changing the freezing point of water molecules. This process suppresses the growth of ice crystals, resulting in smaller ice crystal volumes, which reduces damage to muscle tissue and helps maintain the structural integrity of proteins, thereby minimizing the deterioration of meat texture. The combined application of electric fields and superchilling in meat significantly enhances the supercooling effect, inhibits the nucleation of water molecules, and leads to smaller and more uniformly distributed ice crystals formed during superchilling, effectively improving the quality of meat after thawing. When an electric field is applied to superchilled meat, it alters the transmembrane potential difference of the meat cells, leading to the directed movement of electric charges across the membrane, which creates an electric current that disrupts normal physiological metabolism within the cells. This disruption inhibits the growth and reproduction of microorganisms and reduces the free energy required for ice crystal nucleation, thereby increasing the rate of ice formation and minimizing mechanical damage to cellular structures [60]. Research by Xiang et al. [61] demonstrated that when low-pressure static electric fields are combined with superchilling for the storage of small yellow croakers, the degree of denaturation of salt-soluble proteins is significantly reduced, effectively controlling bacterial proliferation and decreasing the occurrence of fissures in muscle tissue caused by mechanical damage from ice crystals.

### 4.2. Composite Preservation Application of Superchilling and Magnetic Fields

Magnetic fields commonly applied in food include static magnetic fields, alternating magnetic fields, and oscillating magnetic fields. Static magnetic fields are generated by direct current, maintaining a constant magnetic field strength and direction. Alternating magnetic fields are produced by alternating current, resulting in regular variations in both magnetic field strength and direction. Oscillating magnetic fields are generated by oscillating currents, which are a type of alternating current with a higher frequency. Regardless of whether they are static, alternating, or oscillating, all magnetic fields can influence the nucleation process of water by rotating or vibrating water molecules and altering the structure of water clusters. This leads to changes in the arrangement of water molecules, increasing the supercooling of water, allowing it to remain in a liquid state even below its freezing point, while also accelerating the freezing rate of water molecules, resulting in the formation of smaller and more uniform ice crystals [62]. He et al. [63] found that when the static magnetic field strength is 8 mT, pork can be stored under superchilling conditions without ice nucleation for 12 days, preventing freeze damage during superchilled storage. Compared to refrigerated, superchilled, and frozen treatments, the juice loss rate of pork subjected to magnetic field-assisted superchilling was reduced by 21.10%, 59.81%, and 30.64%, respectively. Additionally, the indicators of color, tenderness, centrifuge loss, myofibrillar protein solubility, and surface hydrophobicity were optimal, while the changes in pH and lipid oxidation were the slowest. This treatment also reduced the total bacterial count, indicating that magnetic field-assisted superchilled storage can effectively maintain meat freshness and water retention. However, there is relatively little exploration of the key magnetic field strength parameters for the composite preservation application of superchilling and magnetic fields, necessitating further research. Lee et al. [64] discovered that salmon stored under superchilling conditions with oscillating magnetic field treatment showed no significant difference in color compared to fresh salmon. Moreover, it exhibited lower drip loss, lower TBARS values, and fewer total bacteria counts compared to refrigerated and frozen samples, effectively preserving the quality of the salmon.

### 4.3. Composite Preservation Application of Superchilling and Electron Beams

Electron beam radiation employs low-energy or high-energy pulsed electron beams generated by electron accelerators, which can directly penetrate food to induce physical, chemical, and biological changes, thereby achieving antimicrobial and sterilization effects [65]. Currently, low-energy electron beams with energy levels below 0.3 MeV are primarily utilized in food sterilization. The mechanisms of low-energy electron beam sterilization and preservation can be categorized into two main effects: direct effects and indirect effects. The direct effect refers to the penetration of low-energy electron beams into food, which damages the DNA of microbial cells, leading to the degradation of DNA bases or the breakage of hydrogen bonds, resulting in the denaturation of proteins and enzymes and the disruption of microbial metabolism, ultimately causing cellular injury or death [66]. The indirect effect involves the radiation-induced decomposition of water molecules, generating free radicals that participate in redox reactions and interact with physiologically active substances in microorganisms, leading to structural damage and effectively killing pathogenic microorganisms and other spoilage bacteria in food [67]. The combined use of electron beam irradiation and superchilling further inhibits microbial activity, causing damage to microbial DNA and cellular structures, thereby effectively inactivating pathogenic and spoilage bacteria. Moreover, at low temperatures, electron beam treatment has minimal impact on meat protein denaturation and texture. Yang et al. [68] demonstrated that storage with low-energy electron beams (0.2 MeV; 8 kGy) combined with superchilling can increase the a* value and extend acceptable sensory characteristics to 30 days, significantly reducing the total bacterial count and lowering the TVB-N value, thereby better maintaining meat quality. Furthermore, this approach successfully eliminated Listeria, Bacillus, and Lactobacillus, thus extending the shelf life of pork.

## 5. Conclusions

This paper reviews the current applications of superchilling technology for meat preservation, the principles of preservation, and its combined preservation applications with new technologies. Superchilling, as a preservation method that integrates the advantages of refrigeration and low-temperature freezing (<−8 °C), offers unique advantages and significant potential for regulating meat quality and preservation. By minimizing both the nucleation and growth of ice crystals, reducing the mechanical damage that ice crystals cause to meat, and inhibiting the activity of endogenous enzymes and microorganisms, superchilling technology can effectively reduce moisture loss, maintain desirable tenderness and flavor, control the oxidation of proteins and lipids, slow down microbial growth, and preserve freshness. However, the exclusive use of superchilling technology has certain limitations. The combined application of new technologies, such as electric fields, magnetic fields, and electron beam radiation, with superchilling has demonstrated improved preservation effects in controlling the nucleation and growth rates of ice crystals. Nevertheless, superchilling compound technology is still subject to higher operating requirements and higher costs. With the further popularization of the technology and a decrease of cost input, its application in meat storage has great potential. The future development of superchilling technology and its composite technologies faces many challenges, including the optimization of superchilling techniques, precise control over superchilling temperatures, and the need for specific superchilling temperatures for different types of meat. Superchilling compound technology should break through the limitations of its commercial application to reduce energy consumption and increase production capacity. Further research is needed to provide more scientifically based storage methods and technical support for meat storage and distribution.

## Figures and Tables

**Figure 1 foods-13-03309-f001:**
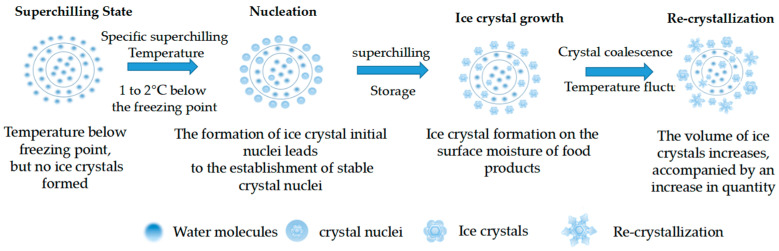
Process of ice crystal formation in superchilling meat.

**Figure 2 foods-13-03309-f002:**
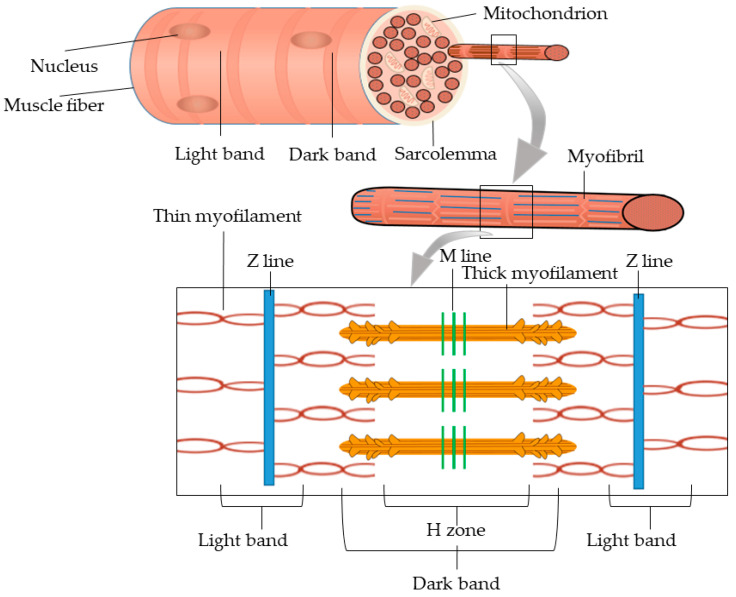
Structure of muscle.

**Figure 3 foods-13-03309-f003:**
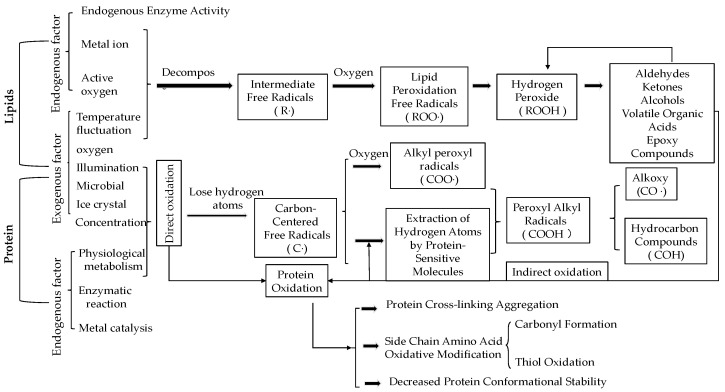
Mechanisms and outcomes of lipid and protein oxidation during superchilling storage of meat.

**Table 1 foods-13-03309-t001:** Research on the application of superchilling for meat preservation.

Storage Object	Superchilling Temperature	Control Temperature	Preservation Effect of Superchilling	References
pork	−2.5 °C	3 °C, −18 °C	Lower drip loss and TVB-N values, stable color, and total bacterial counts during storage that are lower than those in the refrigeration group.	[11]
chicken breast meat	−2.5 °C	3 °C, −18 °C	Significantly higher water holding capacity, color stability, and lower total colony and TVB-N values.	[11]
mutton	−4 °C	4 °C	The decline in pH levels, the consumption of glucose, and the post-mortem glycolytic process have been effectively delayed.	[12]
rabbit	−4 °C	4 °C, −18 °C	Compared to 4 °C, there are lower total aerobic counts, TVB-N values, TBARS values, and pH, while maintaining better tissue structure than at −18 °C.	[13]
beef	−2.1 °C	4 °C	Fewer total bacterial counts, a slower oxidation process of myoglobin, and better preservation of color and water-holding capacity.	[14]
beef	−2 °C	3 °C, −18 °C	Compared to 3 °C, there are lower total aerobic bacteria, pH, and TVB-N values, along with more stable color, while maintaining better tissue structure than at −18 °C.	[15]
duck	−1.66 °C	4 °C	Fewer total bacterial counts, a slower decline in pH, and the maintenance of favorable sensory characteristics.	[16]
micropterus salmoides	−2 °C	4 °C	Lower TVB-N values and fewer total bacterial counts, along with a slower decrease in actin content.	[17]
tuna	−3.2 °C	4 °C, −18 °C	Compared to 4 °C, it exhibits a lower K value, and compared to −18 °C, it maintains better tissue structure and water-holding capacity.	[18]
crayfish	−3 °C	0 °C, 3.5 °C	Lower TVB-N values and fewer colony counts.	[19]

## Data Availability

No new data were created or analyzed in this study. Data sharing is not applicable to this article.

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
