# Peer review of "Research Progress on the Effect and Mechanism of Superchilling Preservation Technology on Meat Quality Control"

_foods, 2024, doi:10.3390/foods13203309_

Round 1
Reviewer 1 Report
Comments and Suggestions for Authors
Please see attached file for comments/questions.

minor issues detected
Reviewer 2 Report
Comments and Suggestions for Authors
Excellent paper regarding the use of super chilling in meats. Comments below:
Line 26. Please review the Keywords to ensure they are useful in attracting people interested in this work. Perhaps name one of the new technologies.
Line 39. "low temperature freezing". All freezing is low temperature. This wording is a bit imprecise. It is used in a couple other places in the paper as well. I think I know what you mean by it, but am not sure everyone else is.
Line 79. The authors write that "the mechanical damage caused by superchilling is lower than . . ." I think they mean that it is less than that of low temperature freezing.
Line 89. Not all readers will know what some of these acronyms mean.
Line 174 ff. The authors write that the water-holding capacity of meat is mainly influenced by two factors. I think what follows is true in frozen meat. There are many other factors that influence the water-holding capacity of meat that are beyond the scope of this paper.
Line 232. The authors should also mention the effects of cold shortening which toughen the meat immensely.
Line 244. The word "maturation" is used more in other contexts by meat scientists. For example, hard salami has a maturation time. However, in current context, the term is not widely used. Perhaps choose a different word or define this one.
Line 332 ff. Please reread this sentence. I am not sure what "metabolic abundance in lamb meat" is being referred to. The context is spoilage. Perhaps there is a word missing.
Round 2
Reviewer 2 Report
Comments and Suggestions for Authors
Nice job with the revisions. A couple of comments below.
Line 62. Perhaps something like this, "During the superchilling process, only a portion of the total water in the meat is frozen -- the content of the generated ice crystals is about 5%-30%."
Line 77 ff. This is a bit awkward. It is a very long sentence. Perhaps making it into two will help with clarity. Perhaps something like this, "Although the shelf life of superchilled meat is far less than that of low-temperature freezing (<-8C), the mechanical damage caused by superchilling is less than that of low-temperature freezing. Consequently, the mechanical damage to muscle/meat structure caused by ice crystal is less than that obtained from low-temperature freezing, which helps reduce the moisture loss during storage." The authors can all the bit regarding freshness if desired. (Note that I could not include the degree sign above).
Line 84. The authors can remove the second TVB-N in the parentheses.
Author Response
Dear Editors and Reviewers:
Thank you for your letter and for the reviewers’ comments concerning our manuscript entitled “Research progress on the effect and mechanism of super-chilling preservation technology on meat quality control”. Thanks to the experts for reviewing our article and giving us honest advice to improve the manuscript. Thank you very much for giving us the opportunity to rework our work and improve the manuscript. You are right that much revision and clarification need to be done in order to paper might possibly be published. Thanks again.
We consider the comments of all reviewers very carefully and revise the manuscript according to the reviewers comments one by one.
The manuscript has been revised carefully according to the reviewers' comments. In our rebuttal indicate the line number in the revised manuscript corresponding to each change that has been made and use yellow highlighting in the text to indicate the edits. We have carefully made correction which we hope to meet with approval. Changes are marked in red in the manuscript. The responds to the reviewers’ comments and all corrections in the paper are as follow:
Response to Reviewer #2
Thank you very much for the recognition of this study. We would like to express our most sincere appreciation for your careful review of this manuscript. The valuable suggestions you had proposed were beneficial to improve the quality of this manuscript. The modifications have been performed carefully according to your suggestions as follow:
Q1: line63. Perhaps something like this, "During the superchilling process, only a portion of the total water in the meat is frozen -- the content of the generated ice crystals is about 5%-30%."
A1: Thank you for correcting the manuscript. In the manuscript, the word has been replaced in line 63-64 as follow:
During the superchilling process, only a portion of the total water in the meat is frozen - the content of the generated ice crystals is about 5% - 30% [7].
Q2: line 72-77. This is a bit awkward. It is a very long sentence. Perhaps making it into two will help with clarity. Perhaps something like this, "Although the shelf life of superchilled meat is far less than that of low-temperature freezing (<-8C), the mechanical damage caused by superchilling is less than that of low-temperature freezing. Consequently, the mechanical damage to muscle/meat structure caused by ice crystal is less than that obtained from low-temperature freezing, which helps reduce the moisture loss during storage." The authors can all the bit regarding freshness if desired. (Note that I could not include the degree sign above).
A2: Thanks for your suggestion. You are right. In the revised manuscript, the sentence has been rewritten in line 72-76 as follow:
Although the shelf life of superchilled meat is far less than that of low-temperature freezing ( < -8C ), the mechanical damage caused by superchilling is less than that of low-temperature freezing. Consequently, the mechanical damage to muscle/meat structure caused by ice crystal is less than that obtained from low-temperature freezing ( < -8℃ ), which helps reduce the moisture loss during storage.
Q3: line 85-86. The authors can remove the second TVB-N in the parentheses
A3: Thanks for your suggestion. In the revised manuscript, the sentence has been rewritten in line 84-85 as follow:
The application effects of superchilling technology in meat preservation are primarily evaluated through indicators such as pH, TBARS (Thiobarbituric acid reactive substance), TVB-N (Total volatile basic nitrogen), K value, and total bacterial count, as shown in Table 1.
Please see the attachment.
